# Is Hemopexin a Nephrotoxin or a Marker of Kidney Injury in Renal Ischemia-Reperfusion?

**DOI:** 10.3390/biom14121522

**Published:** 2024-11-27

**Authors:** You Hyun Jeon, Eun-Joo Oh, Se-Hyun Oh, Jeong-Hoon Lim, Hee-Yeon Jung, Ji-Young Choi, Jang-Hee Cho, Sun-Hee Park, Yong-Lim Kim, Chan-Duck Kim

**Affiliations:** 1Division of Nephrology, Department of Internal Medicine, School of Medicine, Kyungpook National University Hospital, Kyungpook National University, Daegu 41944, Republic of Korea; bon2bon@naver.com (Y.H.J.); oej1124@naver.com (E.-J.O.); sehyun.oh@knu.ac.kr (S.-H.O.); jh-lim@knu.ac.kr (J.-H.L.); hy-jung@knu.ac.kr (H.-Y.J.); jyss1002@hanmail.net (J.-Y.C.); jh-cho@knu.ac.kr (J.-H.C.); sh-park@knu.ac.kr (S.-H.P.); ylkim@knu.ac.kr (Y.-L.K.); 2Cell and Matrix Research Institute, Kyungpook National University, Daegu 41944, Republic of Korea

**Keywords:** acute kidney injury, biomarker, hemopexin, ischemia-reperfusion injury

## Abstract

Destabilization of heme proteins is recognized to play a role in acute kidney injury (AKI). Hemopexin (Hpx), known for its role in binding heme, mitigates free heme toxicity. Despite this, the potential adverse effects of Hpx deposition in kidney tissues and its impact on kidney function are not fully understood. Deferoxamine (DFO) chelates iron released from heme and mitigates associated kidney damage. Therefore, this study aimed to evaluate whether Hpx contributes to kidney injury in an ischemia-reperfusion injury (IRI) induced AKI model and to investigate if DFO could alleviate this damage. Mice were categorized into five groups: Sham-Vehicle, Sham-Hpx, IRI-Vehicle, IRI-Hpx, and IRI-Hpx-DFO. Decline in kidney function was observed exclusively in the IRI group, independent of Hpx injection. Serum Hpx levels remained comparable across all groups, and administration of Hpx did not alter serum Hpx levels or kidney function after 24 hours. Although increased Hpx deposition in kidneys was noted in both the IRI and Hpx groups, this accumulation did not correlate with impaired kidney function. Additionally, DFO did not exhibit a protective effect against kidney injury. In summary, Hpx does not directly induce kidney injury and cannot be considered a biomarker for kidney damage caused by IRI.

## 1. Introduction

The incidence of acute kidney injury (AKI) is rising globally, accompanied by high mortality rates among those affected [1,2]. Survivors of AKI often experience persistent and significant impairment of kidney function [3]. AKI is also a contributing factor to the increasing incidence of chronic kidney disease [4]. Despite its unfavorable short- and long-term consequences, established diagnostic and therapeutic advancements are lacking. Current research focuses on identifying biomarkers that can facilitate the early detection of AKI. Addressing these challenges is essential to improving patient care and reducing the global burden of kidney disease.

The current diagnosis of AKI relies on serum creatinine (Cr) levels and urine output [5]. While these markers are widely utilized owing to their cost-effectiveness and ease of measurement, they have limitations in the early detection of AKI. Serum Cr, a surrogate marker of kidney function, often elevates only after kidney injury has occurred rather than directly indicating kidney damage [6]. Recognizing these limitations, researchers are striving to identify novel diagnostic markers for AKI. Biomarkers of kidney damage, such as Neutrophil gelatinase-associated lipocalin (NGAL) [7] and kidney injury molecule-1(KIM-1) [8], have shown promise. However, their clinical application and effectiveness in routine practice require further evaluation and validation [9].

AKI is a heterogeneous clinical syndrome characterized by varied etiologies and pathophysiological mechanisms that collectively cause a rapid deterioration of kidney function [9,10]. Current research into understanding AKI emphasizes its heterogeneity in etiology, clinical phenotype, and underlying pathology [2]. Specific types of kidney injury increase the levels of free heme and heme-binding proteins that filter into the kidneys [11]. These mechanisms encompass oxidative stress, cytotoxicity, and activation of proinflammatory processes [11]. Heme toxicity is a primary feature of AKI associated with hemolysis and rhabdomyolysis and is also recognized in conditions involving inflammatory and immune responses [12]. The influence of heme-driven pathology in AKI extends beyond hemolysis and rhabdomyolysis, underscoring its significant role in the broader pathophysiology of AKI [13]. Notably, heme content is also significantly increased in AKI not directly associated with heme proteins, such as IRI-induced and cisplatin-induced kidney injury [14,15].

Heme is an iron-containing molecule consisting of four pyrrole rings, and it is biosynthesized predominantly in the liver and bone marrow [16]. As an essential component of hemoglobin, as well as myoglobin and cytochromes, heme plays critical roles in biological systems, including oxygen transport and the catalysis of various biochemical reactions. Heme homeostasis is maintained through stringent regulation of the heme synthesis, recycling, and degradation. Senescent red blood cells undergo degradation in macrophage in the spleen. Heme Oxygenase (HO), which converts heme to biliverdin, is essential in the heme catabolic process [17]. Due to its antioxidant properties, it has been demonstrated to confer protective effects against heme-driven AKI [18,19].

When heme homeostasis is disrupted, excess heme exhibits toxic properties. Unbounded heme becomes highly cytotoxic, promoting free radical production, protein oxidation, lipid peroxidation, and DNA damage [20]. The lipophilic nature of free heme disrupts mitochondrial and nuclear lipid bilayers and sensitizes cells to programmed cell death in response to stimuli [21]. In addition, an increase in heme burden can lead to iron overload. In redox reactions, the iron acts as an electron sink; however, when present in excess, it can exacerbate tissue damage by promoting oxidative injury to cellular membranes and proteins [22]. Deferoxamine (DFO), is used pharmacologically to treat iron overload in patients with conditions such as β-thalassemia major and hemochromatosis by chelating excess iron through its high-affinity hydroxamic groups [23]. It is also shown to be protective in various AKI models associated with increased heme burden [24,25].

Hemopexin (Hpx), a heme-binding glycoprotein, plays a crucial role in scavenging released extracellular heme [26]. Hpx binds strongly to heme in plasma, thus protecting tissues from heme-induced oxidative stress [26]. This protective role is supported by studies showing that AKI occurs predominantly in Hpx knockout mice under hemolysis stress [27]. However, other studies report that the protease activity of Hpx can disrupt the glomerular filtration barrier, leading to proteinuria [28,29]. Additionally, recent studies show that Hpx accumulates in the proximal tubules and glomeruli, exacerbating kidney damage in murine AKI models [30]. However, the precise role of Hpx in kidney disease remains unclear. Therefore, this study aims to determine whether Hpx induces kidney injury in an ischemia-reperfusion injury (IRI)-induced AKI model. We also explored the protective effect of DFO on hemoglobin-induced injury.

## 2. Materials and Methods

### 2.1. Experimental Design

Eight-week-old male C57BL/6 mice weighing 22–25 g (Samtako, Osan, Korea) were used in this study. All procedures were performed in accordance with the guidelines provided by the Animal Care and Use Committee of Kyungpook National University (KNU-2003-0229). The mice were randomly assigned to five groups: sham operation with vehicle (SV, *n* = 4), sham operation with 2 mg of Hpx per mouse (SH, *n* = 2), IRI-vehicle (IV, *n* = 4), IRI treated with 2 mg of Hex per mouse (IH, *n* = 4), and IH administered intraperitoneal injection of DFO (IHD, *n* = 4). Based on previous studies [31], we used approximately 100 mg/kg of Hpx and preliminary experiments confirmed that both control and IRI groups were well tolerated at a concentration of 50 mg/kg. For ischemia induction, the mice were anesthetized using isoflurane inhalation and kidney pedicles were completely occluded for 30 min using a microaneurysm clamp. After 30 min of ischemia, the artery clamp was removed to allow reperfusion, and the skin was closed. Identical surgical treatment was performed on sham-operated animals except for the clamping of the kidney pedicles. During the operation, animals were maintained at a temperature of 36.5–37 °C using a temperature-controlled heating device (Harvard Bioscience, Holliston, MA, USA). Purified human Hpx (Athens Research, Athens, GA, USA) was dissolved in normal saline. The SV and IV groups (controls) were administered a single vehicle dose of the vehicle only. The drug-treated groups were administered a single vehicle dose with Hpx via the tail vein prior to bilateral IRI. DFO (D9533-1G; Sigma–Aldrich, St. Louis, MO, USA) was injected intraperitoneally once at a dose of 200 mg/kg, following prior research demonstrating its protective effects in IRI models [30,32]. Blood and kidney tissues were collected 24 h after reperfusion.

### 2.2. Kidney Function and Histopathological Studies

Blood urea nitrogen (BUN) and Cr levels in mouse serum were measured by GCLabs (Yongin, Korea) using the Cobas 8000 modular analyzer system (Roche, Mannheim, Germany). Kidney tissues from each experimental group were immersion-fixed in 4% paraformaldehyde (pH 7.4), embedded in paraffin, and sectioned into two-micrometer slices. These sections were prepared and stained with periodic acid-Schiff (PAS) and Masson’s trichrome, following standard protocols to determine histological changes and collagen deposition, respectively. A pathologist, blinded to the group information, assessed the slides using the following qualitative scoring system: 0 = no renal injury; 1 = slight renal injury with ≤25% area affected; 2 = moderate injury with 26% to 50% area affected; 3 = significant renal injury with 51% to 75% area affected; and 4 = severe renal injury with 75% to 100% area affected. Ten ×20 fields per kidney were evaluated.

### 2.3. Immunohistochemistry

Kidney tissues from each experimental group were immersion-fixed in 4% paraformaldehyde (pH 7.4) and then embedded in paraffin for sectioning. Immunohistochemical (IHC) staining was performed on 2-μm sections using an anti-Hpx antibody (1:100, 1M603, Novusbio) overnight at 4 °C. Horseradish-peroxidase-conjugated polyclonal goat anti-rabbit immunoglobulin (Dako, Glostrup, Denmark) was used as the secondary antibody. The DAB Peroxidase Substrate Kit (Vector Laboratories, Burlingame, CA, USA) was utilized to visualize the stained tissues. All sections were counterstained with Mayer’s hematoxylin. The immunolabeling was examined under a Leica microscope (Leica Microsystems, Wetzlar, Germany), and the Hpx-positive areas were quantified using the Image J software v.1.54k.

### 2.4. Western Blot Analysis

Immunoblotting of kidney tissues was performed using 10% SDS-polyacrylamide gel electrophoresis with 20 µg of protein, which was then transferred to a nitrocellulose membrane. The membranes were blocked with 10% skimmed milk for 1 hour at room temperature, incubated overnight at 4 °C with primary antibodies against anti-Fibronectin (1:1000, Ab2413, Abcam, Cambridge, UK), KIM-1 (1:1000, PA1-86790, Invitrogen), NGAL (1:1000, Ab216462, Abcam), anti-HO-1(1:1000, ADI-OSA-110, Enzo Life Science, Farmingdale, NY, USA), and anti-glyceraldehyde-3-phosphate dehydrogenase (GAPDH) (1:5000, 2118S, Cell Signaling, Danvers, MA, USA). The membrane was then incubated for 1 h at room temperature with horseradish peroxidase-conjugated secondary antibodies (1:2000; Dako, Glostup, Denmark). Detection was performed using advanced ECL reagents (Amersham Bioscience, Piscataway, NJ, USA) and an enhanced chemiluminescence detection system (GE Healthcare, Buckinghamshire, UK). Band densities were quantified via densitometry and compared to the expression of GAPDH. Band pixel intensities were analyzed using Scion Image software v.4.0 (Scion, Frederick, MD, USA).

### 2.5. Enzyme Linked Immunosorbent Assay

The samples were collected in tubes with serum-separating agent and then centrifuged for 10 min at 3000 rpm at 4 °C. Then, the serum samples were immediately frozen and stored at −80 °C until analysis. Serum Hpx (Abcam, Cambridge, UK) and NGAL (R&D Systems, Minneapolis, MN, USA) concentrations were measured using a commercially available enzyme-linked immunosorbent assay kit according to the manufacturer’s instructions. Values were calculated using a standard curve.

### 2.6. Statistical Analysis

The data are presented as means ± standard error of the mean. The experiments were repeated independently at least three times. The statistical analyses were conducted using a one-way analysis of variance followed by Tukey’s post hoc test performed with GraphPad Prism v.5.01 software (GraphPad Software, La Jolla, CA, USA). *p* < 0.05 was considered statistically significant.

## 3. Results

### 3.1. Kidney Function and Renal Injury Scores

To evaluate the effect of Hpx on kidney function, we analyzed serum Cr and BUN levels across different groups. Serum Cr and BUN levels were elevated in the IRI groups compared to those in the sham groups (all *p* < 0.05; Figure 1A,B). No significant differences in kidney function were observed between the SV and SH groups, regardless of Hpx infusion (Figure 1A,B). Similarly, kidney function remained constant between the IV and IH groups (Figure 1A,B). In the IHD group, kidney function was comparable to that in the IV and IH groups (Figure 1A,B). Histological analysis of renal injury scores revealed that IV, IH, and IHD groups exhibited significantly higher scores than those of the SV and SH groups (Figure 1C,D). These results reveal that Hpx infusion, regardless of IRI, does not alter kidney function.

### 3.2. Increased NGAL Expression in IRI Groups Independent of Hpx Infusion

Next, we assessed markers of kidney injury and fibrosis across all groups. Kidney dysfunction induced by IRI correlated with increased expression of NGAL in the kidneys, showing a significant rise in IRI groups compared to that in the sham groups (Figure 2A,C). Administration of Hpx in the absence of IRI did not significantly alter these markers (SV vs. SH, all *p* > 0.05; Figure 2). Furthermore, when Hpx was administered in the presence of IRI, no changes were observed in injury markers (IV vs. IH, all *p* > 0.05; Figure 2). The expression of HO-1 protein was elevated in the IH group compared to that in the SV group (*p* < 0.05; Figure 2E). However, DFO did not significantly mitigate the expressions of NGAL and HO-1 in the IHD group (IH vs. IHD, all *p* > 0.05; Figure 2C,E).

### 3.3. Distinct Serum Profiles of NGAL and Hpx in Response to IRI

To better understand the role of Hpx in kidney damage resulting from IRI, we measured serum levels of the NGAL and Hpx across various experimental groups. Figure 3 illustrates that serum levels of NGAL and Hpx exhibit distinct responses to IRI. Serum NGAL levels significantly increased in the IRI group compared to the sham group (*p* < 0.05, Figure 3A), whereas serum Hpx levels were similar between the sham and IRI groups.

### 3.4. Collagen Deposition in the IRI Groups

The extent of collagen deposition in the kidneys was assessed across the groups, showing significant collagen deposits in the kidney interstitium of both IV and IH groups (all *p <* 0.01, Figure 4). Hpx administration exhibited no discernible effect in the sham or IRI groups, with no significant differences observed (SV vs. SH, IV vs. IH, all *p* > 0.05; Figure 4B). Treatment with DFO showed a trend towards attenuating collagen deposition but did not reach statistical significance (Figure 4B).

### 3.5. Hpx Deposition Induced by IRI and Hpx Infusion

We performed an IHC analysis to compare Hpx deposition in the kidneys across different groups. The SH group showed significantly higher levels of Hpx deposition than that of the SV group, even in the absence of IRI (SV vs. SH, *p* < 0.0001; Figure 5B). Similarly, the IV group showed elevated intrarenal Hpx levels (SV vs. IV, *p* < 0.0001; Figure 5B). However, no significant difference was observed between the two groups (SH vs. IV group, *p* > 0.05; Figure 5A,B). The combination of IRI and Hpx infusion resulted in increased Hpx deposition compared to that when IRI was employed alone (IV vs. IH, *p* < 0.05; Figure 5B). This enhancement cannot be alleviated by treatment with DFO (IH vs. IHD, *p* > 0.05; Figure 5B).

## 4. Discussion

The present study aimed to evaluate the potential nephrotoxic effects of Hpx and revealed that Hpx does not cause kidney injury. Administration of Hpx had no effect on either serum Hpx levels or kidney function after 24 h, and Hpx infusion did not exacerbate kidney injury caused by IRI. Although the IRI and SH groups showed increased Hpx deposition in the kidneys, this accumulation did not correlate with a decline in kidney function. Additionally, DFO did not exhibit a significant protective effect in the IRI model used in this study. Based on the data presented, Hpx is neither a nephrotoxin nor could it be considered a marker for kidney damage in the context of IRI.

Previous research primarily focused on the toxicity of heme on AKI, particularly in cases of hemolysis and rhabdomyolysis [11,33,34]. However, other studies also highlight increased heme levels in AKI models induced by cisplatin [15]. IRI is also known to elevate heme levels in the kidneys of animal models of AKI [14]. This consistent increase in heme across various AKI etiologies suggests a potential convergent pathway in AKI pathogenesis driven by heme-related mechanisms. Furthermore, recent advances have expanded the understanding of the harmful effects of heme on AKI to include not only free heme but also heme proteins [11,13,35]. These proteins, when destabilized under pathological conditions, can release heme, which is intrinsically associated with kidney injury [36,37,38].

Hpx, a heme-binding protein mainly synthesized in hepatocytes, plays a crucial role in clearing extracellular heme [26]. Beyond heme scavenging, Hpx possesses antioxidant properties, providing protection against heme-mediated injury. Extensive preclinical studies highlight the beneficial effects of Hpx, especially in diseases involving intravascular hemolysis, such as sickle cell disease [39,40,41]. Studies by Ofori-Acquah et al. provide compelling evidence supporting the protective role of Hpx in AKI associated with sickle cell disease. Their research indicates that infusion of Hpx mitigates AKI in mice with sickle cell, while Hpx deficiency worsens kidney dysfunction [27]. Another study reveals that Hpx prevents kidney damage by inhibiting the activation of complement component C3 in hemolytic conditions [31]. In experimental models of AKI induced by rhabdomyolysis, Hpx also demonstrates protective effects by attenuating complement activation and subsequent kidney damage [34]. These findings collectively underscore the protective roles of Hpx in mitigating kidney injury in heme-driven pathologic conditions linked to AKI.

Nevertheless, the precise role of Hpx in kidney disease remains unclear. Hpx is implicated as a permeability factor in nephrotic syndrome [42], suggesting its involvement in the pathophysiology of glomerular dysfunction. Experimental evidence demonstrates that purified human plasma Hpx can induce proteinuria in rats by disrupting the glomerular filtration barrier [29]. Studies on children with relapsed minimal change disease report elevated Hpx levels compared to those in remission [43]. Moreover, the protease properties of Hpx are selectively activated under certain conditions, potentially triggering podocyte remodeling and affecting the permeability dynamics of the glomerular filtration barrier [42].

Our study revealed that Hpx administration did not lead to an increase in kidney injury markers or histological changes, indicating that Hpx alone does not appear to be nephrotoxic. IHC analysis revealed increased Hpx deposition in the kidneys of the IRI groups, as well as in the group receiving Hpx injection without IRI. Infusing Hpx with IRI further increased this deposition in the kidneys, but in the absence of IRI, kidney function remained unaffected. These findings suggest that the deposition of Hpx in the kidneys may not be a part of the mechanism of kidney injury induced by IRI. The increased Hpx deposition observed in the SH group may be the result of enhanced glomerular filtration and tubular uptake of the administered Hpx, while in the IRI group, it may suggest delayed excretion of Hpx due to reduced kidney function at 24 h time point. Fan et al. demonstrate that coincubation of Hpx and hemoglobin increased ROS, apoptosis, and necrosis in HK2 cells; nonetheless, they observed that Hpx alone did not induce cellular injury [30]. These results are consistent with our results, suggesting that while Hpx does not possess intrinsic nephrotoxic properties its interaction with hemoglobin may contribute to cellular damage. Our findings show similar serum Hpx levels between the sham and IRI groups, consistent with prior research showing that serum Hpx levels in the sham group were elevated compared to the control group, while similar to those observed in the AKI group [44]. This suggests that Hpx acts as an acute phase reactant in response to various stressors. However, since serum levels did not align with the extent of renal deposition, it is likely that renal Hpx accumulation is driven by mechanisms independent of serum Hpx concentrations.

Studies using DFO demonstrated its effectiveness in reducing oxidative damage and providing protection in hemoglobin- and myoglobin- induced AKI models [24,25]. However, the current study found that DFO did not confer a protective effect against kidney damage induced by IRI. Several factors may account for this discrepancy. IRI involves complex processes, including inflammation and oxidative stress [45], which may not be directly modifiable through iron chelation alone. Other pathogenic mechanisms could potentially obscure the benefits of iron chelation in the context of IRI. In addition, unlike conditions such as hemolysis or rhabdomyolysis, where heme accumulation results directly from the release of hemoglobin or myoglobin from damaged cells, heme elevation in IRI primarily arises from intracellular cytochrome P450 [14]. This difference in heme source may not produce a level of iron toxicity sufficient to induce significant kidney injury. Collectively, these observations suggest that iron toxicity may not be the primary or sole driver of kidney injury in IRI models, indicating that additional contributing factors are likely involved.

Hpx is studied as a biomarker in various diseases owing to its involvement in diverse pathological mechanisms [46,47,48]. In children with sickle cell anemia, a study focused on changes in the plasma proteome after treatment indicates that Hpx correlates with higher levels of fetal hemoglobin, suggesting its potential utility in monitoring therapeutic responses [49]. Another study on focal segmental glomerulosclerosis found elevated levels of Hpx in patients with steroid-resistant disease, highlighting Hpx as a biomarker for glomerular filter damage [50]. These findings suggest that Hpx may serve as a biomarker for assessing disease severity and monitoring treatment outcomes in intravascular hemolysis and nephrotic syndrome.

However, the role of Hpx as biomarker in the context of the IRI model remains poorly understood. In our study, we found that serum Hpx levels were comparable 24 h after IRI, while the degree of Hpx deposition in the kidneys varied among the groups. This increased deposition of Hpx in the kidneys is likely a consequence of compromised kidney function rather than an indicator of renal damage. Additionally, the administration of DFO did not yield a protective effect against kidney injury. Importantly, at the 24 h time point, neither serum Hpx levels nor Hpx deposition in the kidneys showed any significant association with a decline in kidney function. Collectively, these findings suggest that Hpx is unlikely to play a role in the mechanisms underlying kidney injury in the IRI-induced AKI model. Therefore, Hpx cannot be regarded as a reliable biomarker in this context.

The current study has several limitations. First, as this study was designed to evaluate the nephrotoxic properties of Hpx, additional parameters related to heme metabolism were not investigated. Given the degree of deposition of Hpx and the discrepancy in kidney function, Hpx is unlikely to cause kidney injury directly or initiate the associated mechanisms. Nevertheless, since the accumulation of Hpx in the kidneys leads to exacerbated hemoglobin-induced injury [30], investigating the source of heme in the IRI model may provide deeper insights into the mechanisms of Hpx-mediated renal injury. Furthermore, ferroptosis—a newly identified form of regulated cell death characterized by iron overload and lipid peroxidation—has been closely associated with IRI [51]. Although iron chelation did not protect against IRI in this study, further exploration of the interplay between ferroptosis and Hpx, as well as their collective impact on IRI, is warranted. Additionally, it is worth considering that the level of injury in the IRI model used in this study may have limited the ability to detect potential beneficial effects of DFO treatments. Future studies employing models with varying degrees of injury may help to better elucidate the therapeutic potential of Hpx and related interventions. In addition, we used exclusively male animals to reduce potential confounding effects associated with hormonal fluctuations. It is important to acknowledge the potential influence of sex-related biological differences on the results. While the present study exclusively utilized male mice to reduce potential confounding effects of hormonal fluctuations, future studies that include both male and female animals will be necessary to fully understand the impact of sex on the observed effects. Finally, the relatively small sample size of four animals per group is a limitation of this study and may restrict the generalizability of the findings. Further studies with larger sample sizes are necessary to confirm these observations and enhance the reliability of the conclusions. Recent advances in understanding Hpx-hemoglobin interaction can offer valuable insights for future studies, particularly in elucidating the structural and functional aspects of Hpx-protein interactions and their role in renal pathophysiology [52]. Future research should aim to identify the specific conditions or factors under which Hpx potentially mitigates kidney injury. Addressing these aspects in future research could offer a more comprehensive understanding of the underlying mechanisms of IRI.

## 5. Conclusions

Our results show that Hpx infusion does not significantly alter kidney function in the absence of IRI. In the presence of IRI, additional infusions of Hpx did not cause further worsening of kidney function, indicating that Hpx administration is not directly associated with deteriorated kidney function. Neither the serum Hpx levels nor the degree of Hpx deposition in the kidneys correlated with the decline in kidney function caused by IRI. In conclusion, Hpx itself does not exhibit nephrotoxic effects and cannot serve as a biomarker of kidney damage resulting from IRI.

## Figures and Tables

**Figure 1 biomolecules-14-01522-f001:**
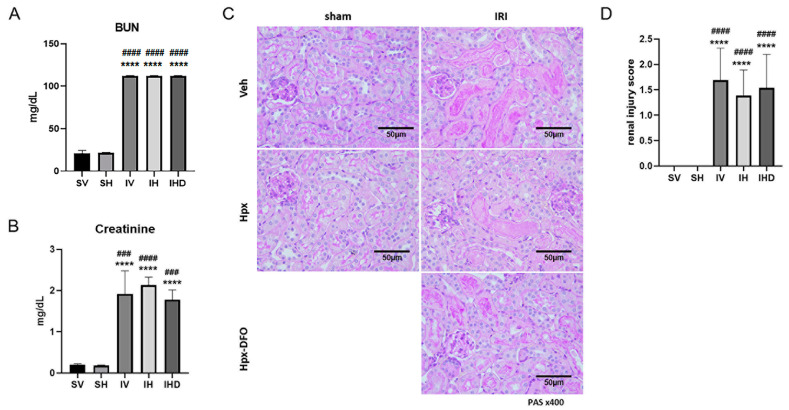
Kidney function was decreased in ischemia-reperfusion injury (IRI) groups. (**A**) Serum blood urea nitrogen levels in each group; (**B**) serum creatinine levels in each group; (**C**) representative microscopic images of kidney tissues stained with periodic acid-Schiff (PAS) (×400). Scale bar = 50 µm; (**D**) Renal injury scores in each group. **** *p <* 0.0001 versus SH group. #### *p* < 0.0001, ### *p* < 0.001 versus SV group.

**Figure 2 biomolecules-14-01522-f002:**
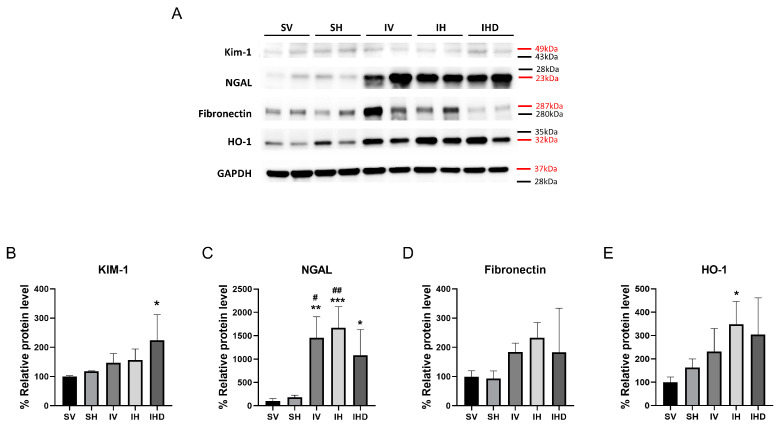
IRI increased Neutrophil Gelatinase-Associated Lipocalin (NGAL) expression in kidneys irrespective of hemopexin (Hpx) infusion. (**A**) Western blot analysis of kidney injury-related markers. Protein levels of (**B**) kidney injury molecule (KIM)-1, (**C**) NGAL, (**D**) fibronectin, and (**E**) Heme oxygenase (HO)-1 determined using Western blot. Red markers: target protein molecular weight, Black markers: protein standards marker. *** *p* < 0.001, ** *p* < 0.01, * *p* < 0.05 versus SV group, ## *p* < 0.01, # *p* < 0.05 versus SH group. Original images can be found in Appendix A, Appendix A.

**Figure 3 biomolecules-14-01522-f003:**
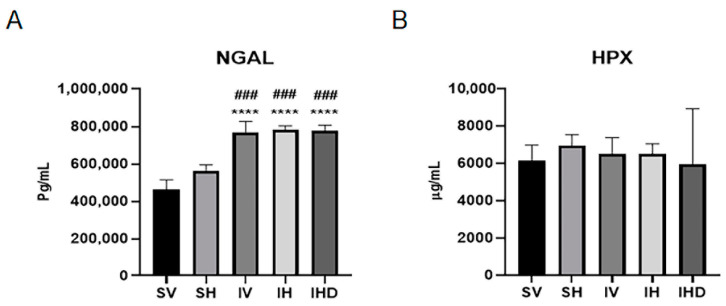
Serum NGAL and Hpx levels exhibit different responses to IRI. Serum levels of (**A**) NGAL and (**B**) Hpx in the SV (*n* = 4), SH (*n* = 2), IV (*n* = 4), IH (*n* = 4), and IHD groups (*n* = 4). **** *p* < 0.0001 versus SV group. ### *p* < 0.001 versus SH group.

**Figure 4 biomolecules-14-01522-f004:**
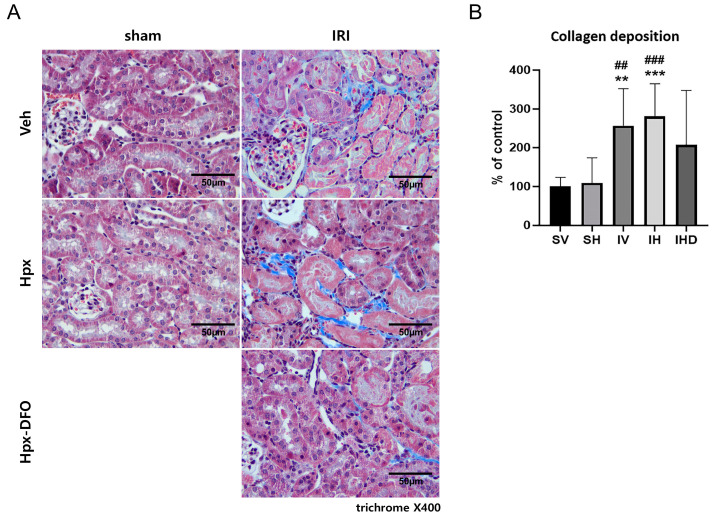
Collagen deposition was observed exclusively in the IRI groups. (**A**) Representative images of kidney tissue stained with Masson’s trichrome (×400), Scale bar = 50 µm; (**B**) Semi-quantitative analysis of collagen deposition area. *** *p <* 0.001, ** *p <* 0.01 versus SV group. ### *p* < 0.001, ## *p* < 0.01 versus SH group.

**Figure 5 biomolecules-14-01522-f005:**
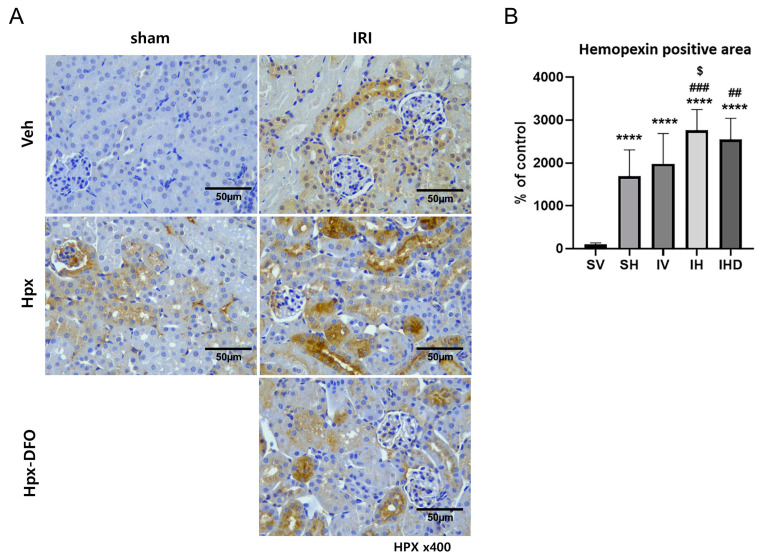
Hpx deposition in response to Hpx infusion and IRI. (**A**) Representative images of Hpx staining in the kidney tissues (×400), Scale bar = 50 µm; (**B**) Semi-quantitative analysis of Hpx positive areas in each group. **** *p <* 0.0001 versus SV group. ### *p* < 0.001, ## *p* < 0.01 versus SH group. $ *p* < 0.05 versus IV group.

## Data Availability

The data presented in this study are available on request from the corresponding author.

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
