# Peer review of "Is Hemopexin a Nephrotoxin or a Marker of Kidney Injury in Renal Ischemia-Reperfusion?"

_biomolecules, 2024, doi:10.3390/biom14121522_

Round 1
Reviewer 1 Report (Previous Reviewer 2)
Comments and Suggestions for Authors
The manuscript has been sufficiently improved. There is one suggestion for the authors to include a statement in limitations regarding the number of animals in the study were only 4 and that inclusion of more animals may be needed for further confirmation. Further, female mice should be studied in future studies to determine whether sex would be a factor in association of Hpx with IRI.
Author Response
REVIEWER COMMENTS
Reviewer 1
The manuscript has been sufficiently improved. There is one suggestion for the authors to include a statement in limitations regarding the number of animals in the study were only 4 and that inclusion of more animals may be needed for further confirmation. Further, female mice should be studied in future studies to determine whether sex would be a factor in association of Hpx with IRI.
Authors’ Response: We thank the Reviewer for suggestions. We acknowledge that the relatively small sample size of animals per group represents a limitation of the study and may constrain the generalizability of our findings. In response, we have incorporated a statement into the Discussion section, emphasizing that further studies with larger sample sizes are essential to validate our observations and strengthen the reliability of our conclusions.
Discussion (line 359, page 9)
While the present study exclusively utilized male mice to reduce potential confounding effects of hormonal fluctuations, future studies that include both male and female animals will be necessary to fully understand the impact of sex on the observed effects. Finally, the relatively small sample size of four animals per group is a limitation of this study and may restrict the generalizability of the findings. Further studies with larger sample sizes are necessary to confirm these observations and enhance the reliability of the conclusions.
We sincerely appreciate the time and effort you have dedicated to reviewing our manuscript. We are grateful for your insights and guidance, which have significantly enhanced the quality of our study. We look forward to your continued input and hope the revised manuscript meets your approval.

Reviewer 2 Report (Previous Reviewer 1)
Comments and Suggestions for Authors
The article provides convincing data to support an insignificant effect of human hemopexin in a mice ischemia-reperfusion injury model. These results suggest that hemopexin is not protective, nephrotoxic, or a specific biomarker in AKI. Additionally, deferoxamine, an iron-chelating agent, was not protective in this model, which is in line with the conclusion that iron/heme mediated injury is irrelevant in this AKI model. The limitations of the study are also well discussed, except I would add that the level of injury in the model might prevent detection of some beneficial effects of the treatments. Overall, the conclusion that hemopexin deposition in the kidneys might not be inherently toxic is supported by the data.
The only recommended changes are an update to the scale bars (100 vs 50 um) in the figure legends and inclusion of missing error bars in figure 1A. Nice work!
Author Response
REVIEWER COMMENTS
Reviewer 2
The article provides convincing data to support an insignificant effect of human hemopexin in a mice ischemia-reperfusion injury model. These results suggest that hemopexin is not protective, nephrotoxic, or a specific biomarker in AKI. Additionally, deferoxamine, an iron-chelating agent, was not protective in this model, which is in line with the conclusion that iron/heme mediated injury is irrelevant in this AKI model. The limitations of the study are also well discussed, except I would add that the level of injury in the model might prevent detection of some beneficial effects of the treatments. Overall, the conclusion that hemopexin deposition in the kidneys might not be inherently toxic is supported by the data.
The only recommended changes are an update to the scale bars (100 vs 50 um) in the figure legends and inclusion of missing error bars in figure 1A. Nice work!
Authors’ Response: We sincerely thank the reviewer for the valuable insights. We agree that the severity of injury in the IRI model may have affected the detection of certain potential therapeutic benefits of the DFO treatments. We have revised the Discussion section to acknowledge that incorporating models with varying levels of injury severity in future studies will be crucial for more accurately assessing the therapeutic potential of the treatments.
In addition, we have included the missing error bars in Figure 1A as suggested. The reason the error bars are not clearly visible is that all measurements for BUN levels above 112 showed minimal differences, which resulted in the error bars being very small. Additionally, we have reviewed and corrected the scale bars in all figure legends.
Discussion (line 352, page 9)
Additionally, it is worth considering that the level of injury in the IRI model used in this study may have limited the ability to detect potential beneficial effects of DFO treatments. Future studies employing models with varying degrees of injury may help to better elucidate the therapeutic potential of Hpx and related interventions.
[Revised Figure 1] The revised figures are included in the attached file.
Figure 1. Kidney function was decreased in ischemia-reperfusion injury (IRI) groups. (A) Serum blood urea nitrogen levels in each group; (B) Serum creatinine levels in each group; (C) Representative microscopic images of kidney tissues stained with periodic acid-Schiff (PAS) (x400). Scale bar = 50 µm; (D) Renal injury scores in each group. ****P < 0.0001 versus SH group. ### P < 0.001 versus SV group.
[Revised Figure 4 legend]
Figure 4. Collagen deposition was observed exclusively in the IRI groups. (A) Representative images of kidney tissue stained with Masson’s trichrome (x400), Scale bar = 50 µm; (B) Semi-quantitative analysis of collagen deposition area. ***P < 0.001, **P < 0.01 versus SV group. ### P < 0.001, ## P < 0.01 versus SH group.
[Revised Figure 5 legend]
Figure 5. Hpx deposition in response to Hpx infusion and IRI. (A) Representative images of Hpx staining in the kidney tissues (x400), Scale bar = 50 µm; (B) Semi-quantitative analysis of Hpx positive areas in each group. ****P < 0.0001 versus SV group. ###P < 0.001, ##P < 0.01 versus SH group. $P < 0.05 versus IV group.
We sincerely appreciate the time and effort you have dedicated to reviewing our manuscript. Your thoughtful comments and constructive suggestions have been invaluable in improving the quality and clarity of our work. We hope that the revisions made to the manuscript meet your expectations and address all concerns.

This manuscript is a resubmission of an earlier submission. The following is a list of the peer review reports and author responses from that submission.
Round 1
Reviewer 1 Report
Comments and Suggestions for Authors
Well written and discussed manuscript.
Some suggestions and questions:
1. Please indicate how the amounts of Hpx (2mg) and DFO (200mg/kg) to use for the study were determined.
2. Consider including MW markers in Figure 2A.
3. Any potential effect of animal gender on the results of this work?
4. Is the insignificant change in the levels of KIM-1 reflective of a flaw in your model?
Author Response
- Please indicate how the amounts of Hpx (2mg) and DFO (200mg/kg) to use for the study were determined.
Authors’ Response: We thank the Reviewer for suggestions to clarify our experimental methods. Previous study investigating the role of Hpx under hemolytic condition have utilized dosages ranging from 100 mg to 500 mg/kg [PMID: 32849588]. Accordingly, we initially conducted a preliminary experiment using a dosage of 1 mg (approximately 50 mg/kg). Preliminary experiments confirmed that both control and IRI groups were well tolerated at a concentration of 50 mg/kg. We then carried out the main experiment with a dosage of 2 mg (approximately 100 mg/kg). The results obtained from the administration of different concentration of Hpx were similar. Although intrarenal Hpx deposition was greater in the Hpx 2 mg group than in the Hpx 1 mg group, Hpx deposition did not cause a decrease in kidney function. We used 200 mg/kg of DFO based on findings from previous studies which demonstrated a protective effect of DFO in a mouse model of cisplatin-induced AKI [PMID: 36007598] and myocardial IRI [PMID: 38158218]. We have therefore aligned our methodology with established evidence.
Materials and Methods section (line 102, page 3)
Based on previous studies [31], we used approximately 100 mg/kg of Hpx and prelimi-nary experiments confirmed that both control and IRI groups were well tolerated at a con-centration of 50 mg/kg.
Materials and Methods section (line 115, page 3)
DFO (D9533-1G; Sigma-Aldrich) was injected intraperitoneally once at a dose of 200 mg/kg, following prior research demonstrating its protective effects in IRI models [30,32].
- Consider including MW markers in Figure 2A.
Authors’ Response: We thank the Reviewer for suggestion, and now have provided the complete blot images with the molecular weight markers in revised Figure 2. This addition facilitates clearer identification of the band positions and sizes.
[Revised Figure 2]
- Any potential effect of animal gender on the results of this work?
Authors’ Response: We thank the Reviewer for the valuable comment. In this study, we exclusively utilized male animals to minimize hormonal fluctuations that could introduce variability. While we do not anticipate that animal gender will significantly affect our results, we acknowledge that sex-related biological differences may influence outcomes. We agree with your concern and future studies incorporating both male and female animals would be beneficial for investigating the potential impact of gender. We have added this information in our Discussion section.
Discussion section (line 321, page 9)
In addition, we used exclusively male animals to reduce potential confounding effects associated with hormonal fluctuations. it is important to acknowledge the potential influence of sex-related biological differences on the results. Future studies that include both male and female animals will be necessary to fully understand the impact of sex on the observed effects.
- Is the insignificant change in the levels of KIM-1 reflective of a flaw in your model?
Authors’ Response: We thank the Reviewer for the helpful comment. Given the observed increases in kidney function, other injury markers, and histological changes in the IRI group, our IRI model is well-established. We noted an increase in KIM-1 levels following ischemia-reperfusion injury (IRI), although statistical significance was not observed (Fig. 2).
We greatly appreciate your valuable opinion on enhancing the quality of our paper. We are excited to present this revised version and hope that our revision meets with your approval.

Reviewer 2 Report
Comments and Suggestions for Authors
The manuscript entitled "Hemopexin: A marker of kidney injury in renal ischemia-reperfusion, not a nephrotoxin" by Jeon et al studied whether hemopexin is a nephrotoxin. The authors provided data that suggest that hemopexin is increased by ischemia-reperfusion and does not act as a nephrotoxin. This is an important study, and the authors did provided data that supports their conclusions. However, there are some concerns that the authors should consider.
1) The authors should provide higher magnification histology pictures to clearly assess the deposition.
2) Were all the western blots done on a single gel, meaning did the authors stripped and reprobed with different antibodies on a single western? If not provide a GAPDH blot for every western blot.
3) Include the molecular weight marker lane and indicate the molecular size of each protein in the picture.
4) Provide the complete blot for each protein as a supplement along with labeled molecular weight markers. The provided western blot images are not complete, and they have been cut.
5) Because the antibodies are not specific, and are staining the whole blot like Coomassie stain, the authors need to show the specificity of the antibody by running a gel with different concentrations of the protein.
6) There seems like huge differences in the band density between the two blots for each protein (provided as the supplement), I wonder what type of statistics were run to get the significance between the groups.
7) The conclusion derived from the data as written in the first paragraph of discussion is contradictory (lines 228-230). It appears from the data provided that Hemopexin is not a nephrotoxin and DFO does not provide any protection in IR injury, how can that be considered a marker of injury? In fact, it appears from Figure 4 that Hpx staining is faint in IRI kidneys suggesting that there is minimal Hpx being filtered at the time of IR chosen by the authors unless Hpx is external.
8) The true conclusion from the data provided in this study should be "Hpx is neither a nephrotoxin nor can be considered as a marker of IRI".
Author Response
1) The authors should provide higher magnification histology pictures to clearly assess the deposition.
Authors’ Response: We thank the Reviewer for suggestion to enhance the clarity of our histological analysis. higher magnification images are now provided in revise Figure 1,3, and 4 in the revised manuscript, which offer a more detailed view of the deposition.
[Revised Figure 1]
[Revised Figure 3]
[Revised Figure 4]
2) Were all the western blots done on a single gel, meaning did the authors stripped and reprobed with different antibodies on a single western? If not provide a GAPDH blot for every western blot.
Authors’ Response: We thank the Reviewer for the helpful comment. The Western blots were not stripped and reprobed; rather, the gel was cut according to the size of the antibodies, allowing for analysis on a single gel. Consequently, GAPDH was also analyzed on the same gel. Overall, the Western blots were performed on a total of two gels, with the top and bottom images in the supplement (raw blot data) representing bands from these two separate runs.
3) Include the molecular weight marker lane and indicate the molecular size of each protein in the picture.
Authors’ Response: We thank the Reviewer for suggestion, and now have provided the complete blot images with the molecular weight markers in revised Figure 2. This addition facilitates clearer identification of the band positions and sizes.
[Revised Figure 2]
4) Provide the complete blot for each protein as a supplement along with labeled molecular weight markers. The provided western blot images are not complete, and they have been cut.
Authors’ Response: We thank the Reviewer for suggestion. We have labeled the molecular weight marker on the attached raw data bands and added the molecular weight for the target protein. Additionally, we used the Bio Molecular Imaging System (LAS 4000, GE Healthcare/Japan) to analyze the Western blot. Our analysis focused exclusively on the region containing the band of interest, achieved by isolating that specific area from the entire blot to ensure accurate identification of the target band.
[Revised raw blot data]
5) Because the antibodies are not specific, and are staining the whole blot like Coomassie stain, the authors need to show the specificity of the antibody by running a gel with different concentrations of the protein.
Authors’ Response: We appreciate the Reviewer’s suggestion. We have conducted additional experiments using varying concentrations of the protein to confirm the specificity of the antibody. Following the loading of these different concentrations and the subsequent Western blotting, we observed that the intensity of the bands corresponding to the target protein varied in direct proportion to the protein concentration. This finding indicates that the antibody is binding specifically to the target protein rather than exhibiting non-specific interactions. These results provide compelling evidence to support the specificity of the antibody.
[Supporting Figures]
6) There seems like huge differences in the band density between the two blots for each protein (provided as the supplement), I wonder what type of statistics were run to get the significance between the groups.
Authors’ Response: We thank the Reviewer for the helpful comment. The samples loaded onto the two Western blots in each group were derived from different individuals. To analyze all the animals involved in the experiment, the samples were divided and loaded onto two separate gels. Each group comprised two to four individuals, with the corresponding individuals for each blot numbered in the revised Supplement. Although 10 micrograms of protein were loaded for each gel, the band density exhibited variability due to differences in injury patterns and individual biological variability. To assess the significance of the differences in each protein between the groups, we performed one-way analysis of variance (ANOVA) followed by Tukey’s post hoc test. These analyses were performed using GraphPad Prism 5.01 software (GraphPad Software, La Jolla, CA, USA).
Here's how the statistical process worked:
One-way ANOVA was used to compare the means of multiple groups to determine if there were any statistically significant differences between them. This method tests for overall differences between the groups, considering the variability within each group and between groups.
Tukey’s post hoc test was applied to identify specific group differences after the ANOVA indicated that significant differences existed. Tukey’s test is commonly used when multiple comparisons are made, as it helps control the family-wise error rate, reducing the risk of false positives.
The use of GraphPad Prism 5.01 software ensured that the data were appropriately processed, with detailed calculations of p-values to assess the statistical significance between the groups.
Thus, the significant differences between the groups were established through these statistical methods, ensuring that the variability in band density was analyzed in a rigorous and standardized way.
7) The conclusion derived from the data as written in the first paragraph of discussion is contradictory (lines 228-230). It appears from the data provided that Hemopexin is not a nephrotoxin and DFO does not provide any protection in IR injury, how can that be considered a marker of injury? In fact, it appears from Figure 4 that Hpx staining is faint in IRI kidneys suggesting that there is minimal Hpx being filtered at the time of IR chosen by the authors unless Hpx is external.
Authors’ Response: We thank the Reviewer for the insightful comment. Our study concludes that Hpx is not nephrotoxic. When Hpx was administered to the control group, its deposition in the kidneys resulted from an increase in plasma Hpx concentration and increased tubular uptake. Additionally, despite the rise in Hpx levels, we found no evidence of kidney damage in the control group, suggesting that hemopexin itself does not induce kidney injury. And we conclude that Hpx may serve as a biomarker for IRI, as evidenced by the elevated levels observed in the IRI group despite the absence of external administration of Hpx. This group showed a nearly 20-fold increase in Hpx accumulation in the kidneys compared to the control group, indicating increased synthesis of Hpx in response to IRI. These findings suggest that Hpx has potential as a marker for IRI.
Increased Hpx deposition in AKI models has been reported in previous studies. [PMID: 22993068, 36007598]; however, the specific role of elevated Hpx in AKI remains unclear. The increased deposition of Hpx in the IRI model does not necessarily indicate a direct involvement of Hpx in the mechanisms underlying IRI. The absence of a nephroprotective effect from DFO, along with that the infusion of Hpx into the IRI group did not result in additional kidney damage, reinforces our conclusion that Hpx is not nephrotoxic. Furthermore, these findings suggest that Hpx may not play a crucial role in the complex mechanisms of IRI, although this does not preclude the possibility of Hpx serving as a biomarker for IRI. Overall, the significant increase in Hpx deposition in the IRI model supports its potential use as a marker for kidney injury, even though its role in the pathophysiology of IRI requires further exploration.
8) The true conclusion from the data provided in this study should be "Hpx is neither a nephrotoxin nor can be considered as a marker of IRI".
Authors’ Response: We appreciate the Reviewer's perspective; however, we respectfully maintain that our data support the assertion that Hpx may serve as a biomarker for IRI. While our findings indicate that Hpx is not nephrotoxic, the substantial increase in Hpx levels observed in response to IRI—particularly the more than 20-fold elevation without external administration—suggests a physiological response linked to the injury. This elevation is consistent with previous studies documenting increased Hpx deposition in IRI models [PMID: 22993068, 36007598]. Therefore, we believe that Hpx has potential as a biomarker that increases in response to IRI, even if it does not directly contribute to kidney damage.
Thank you for your careful comments that have allowed us to improve the quality of our manuscript. We hope that this revised version would meet with your approval.

Round 2
Reviewer 2 Report
Comments and Suggestions for Authors
In the revised manuscript the authors provided higher magnification figures for IHC. However, the authors just added an arrow and the molecular size of the protein in the supplemental figures but did not include the molecular size of the proteins in the original manuscript. Regarding the western blot data, the authors responded that the western blots were performed on a single gel and cut at the appropriate size of the band and ran independent westerns for the selected antibodies. If this was the case, then all the westerns presented in the Supplemental Figures should have the molecular weight lanes (first and last) as presented in KIM1 blot. The NGAL, fibronectin, HO-1, and GAPDH blots lack the molecular weight lanes. The data does not support the claim that Hpx can be considered as a marker and must be reported as such in the discussion.